# The Sensory Mechanisms of Nutrient-Induced GLP-1 Secretion

**DOI:** 10.3390/metabo12050420

**Published:** 2022-05-07

**Authors:** Anna Pii Hjørne, Ida Marie Modvig, Jens Juul Holst

**Affiliations:** 1Department of Biomedical Sciences, Faculty of Health and Medical Sciences, University of Copenhagen, 2200 Copenhagen, Denmark; aphjoerne@sund.ku.dk (A.P.H.); ida.modvig@sund.ku.dk (I.M.M.); 2Novo Nordisk Foundation Center for Basic Metabolic Research, Faculty of Health and Medical Sciences, University of Copenhagen, 2200 Copenhagen, Denmark

**Keywords:** GLP-1, incretins, nutrient sensors, enteroendocrine system, L-cells, glucose homeostasis, gastric bypass, T2DM

## Abstract

The enteroendocrine system of the gut regulates energy homeostasis through the release of hormones. Of the gut-derived hormones, GLP-1 is particularly interesting, as analogs of the hormone have proven to be highly effective for the treatment of type 2 diabetes mellitus and obesity. Observations on increased levels of GLP-1 following gastric bypass surgery have enhanced the interest in endogenous hormone secretion and highlighted the potential of endogenous secretion in therapy. The macronutrients and their digestive products stimulate the secretion of GLP-1 through various mechanisms that we have only begun to understand. From findings obtained from different experimental models, we now have strong indications for a role for both Sodium-Glucose Transporter 1 (SGLT1) and the K^+^_ATP_ channel in carbohydrate-induced GLP-1 secretion. For fat, the free fatty acid receptor FFA1 and the G-protein-coupled receptor GPR119 have been linked to GLP-1 secretion. For proteins, Peptide Transporter 1 (Pept1) and the Calcium-Sensing Receptor (CaSR) are thought to mediate the secretion. However, attempts at clinical application of these mechanisms have been unsuccessful, and more work is needed before we fully understand the mechanisms of nutrient-induced GLP-1 secretion.

## 1. Introduction

### 1.1. The Endocrine Intestine

Energy metabolism is a highly regulated process that depends on the signaling between various systems and organs. Essential for the signaling are the gut-derived hormones secreted by the enteroendocrine cells (EECs) of the gastrointestinal system. The EECs are scattered along the entire intestine, alongside absorptive epithelial cells and mucus-secreting cells. Although the enteroendocrine cells (EECs) account for a minor proportion of the total number of intestinal cells, the enteroendocrine system has been described as the largest endocrine organ of the body [1]. The EECs express an impressive number of hormones involved in everything from digestion, gastric motility, gastric emptying, glucose homeostasis, and appetite regulation. Nutrients and non-nutritious compounds stimulate the release of these hormones directly through sensory transporters and receptors or indirectly through their effects on cellular metabolism. Aside from the endocrine action, recent years of research have revealed a potential role for gut-derived hormones in the local activation of intestinal vagal afferent neurons [2,3,4], which could be important for regulating food intake through direct signaling with the central nervous system. This illustrates the complexity of the enteroendocrine system. Indeed, we have only begun to understand the importance of the system in energy homeostasis. Traditionally, EECs have been characterized based on their main secretory products. According to this classification, gastrin is secreted by G-cells, somatostatin by D-cells, cholecystokinin (CCK) by I-cells, secretin by S-cells, glucose-dependent insulinotropic polypeptide (GIP) by K-cells, and glucagon-like peptide 1 (GLP-1) and peptide YY (PYY) are secreted by L-cells. It is now evident that this classification is overly simplified. Several studies have demonstrated the co-expression of hormones traditionally associated with distinct EECs [5,6,7]. Likewise, the ability of individual EECs to alter their hormonal expression profiles, depending on the signaling gradients, has been demonstrated in vitro [8]. Although there is consensus regarding the flaws of the traditional classification, it is commonly used in the literature, as a more precise alternative is missing.

### 1.2. The Therapeutic Potential of GLP-1 and the Nutrient-Sensing L-Cell

Of all the gut-derived hormones, GLP-1 has possibly been subject to the most interest and attention since it was first identified and described more than 30 years ago [9,10,11,12,13,14,15,16,17,18,19,20,21]. The interest in the hormone emerged from the discovery of the critical role of GLP-1 in glycemic control, where GLP-1 functions as an incretin and thus potentiates the glucose-induced insulin response [16,17,20]. Aside from the crucial role of GLP-1 in glucose homeostasis, the hormone is involved in appetite regulation [22], and it has been demonstrated to slow the rate of gastric emptying [23]. The discovery of GLP-1 receptors on intestinal and pancreatic vagal afferent neurons [2,3,4] revealed a direct route of transmission between the gut, pancreas, and brain. This gut–brain–pancreas relationship is a key element in the role of GLP-1 in regulating eating behavior and highlights the remarkable extent of action of the hormone. Through the combination of these properties, GLP-1 has become important in the pharmacological treatment of type 2 diabetes mellitus [24] and, in more recent years, obesity [25]. Initially, the application of GLP-1 as a therapeutic agent was restricted by its rapid rate of degradation, mediated by the enzyme dipeptidyl peptidase-4 (DPP4) [26]. This challenge was overcome by developing resistant, synthetic GLP-1 receptor analogs (GLP1RAs) and DDP4-inhibitors, which are now routinely used in the management of type 2 diabetes [24].

In recent years, there has been increasing interest in targeting endogenous GLP-1 secretion as an alternative or supplement to the existing GLP1RAs. This idea is based on the evidence of a correlation between increased levels of gut-derived hormones and weight loss, as well as the improved glycemic control seen after gastric bypass surgery [27,28,29,30]. The Roux-en-Y gastric bypass (RYGB) causes sustainable weight loss and rapid remission of type 2 diabetes in the majority of operated patients [27,31]. Targeting endogenous EEC secretion might therefore be expected to create bypass-like hypersecretion of gut hormones for the treatment of obesity and type 2 diabetes without surgery. There are several potential benefits to targeting endogenous GLP-1 secretion as opposed to treatments with GLP-1Ras. First, a part of the appetite- and blood glucose-lowering effect of GLP-1 is thought to involve the activation of local sensory nerves of the gut–brain axis (and probably reaching high local concentrations), whereas GLP-1Ras probably mainly act via leaks in the blood–brain barrier. Additionally, selectively targeting the mechanisms controlling L-cell secretion will result in the release of not only GLP-1 but also co-secreted hormones, namely PYY, oxyntomodulin, and perhaps also CCK, GIP, and neurotensin [5,6,7]. These hormones likewise act to inhibit appetite and are insulinotropic, thereby enhancing the insulinotropic and appetite-reducing effect. Finally, as targeting the endogenous secretion reflects normal physiology more closely, it may be possible to avoid the side effects of GLP1-RAs (nausea, vomiting, and diarrhea) affecting some patients [24].

To utilize natural GLP-1 secretion in therapy, a detailed understanding of the molecular mechanisms underlying its secretion is essential. In rats, pigs, and humans, the tissue concentration of GLP-1 peaks in the distal small intestine [32,33], whereas mice have a higher concentration in the colon [33]. It is well-established that GLP-1 is released after the ingestion of a mixed meal [34], with the different components of the food stimulating secretion through distinct mechanisms. Studies in mice have demonstrated that a mixed meal challenge induces more GLP-1 secretion compared with isocaloric glucose [35]. Thus, the presence of all macronutrients may synergistically enhance GLP-1 secretion. Even though the L-cell density is generally greater in the distal part of the intestinal system [32,33], L-cells of the proximal intestine are presumably more important for nutrient-induced GLP-1 secretion under normal circumstances, as absorption predominantly takes place in the proximal part of the intestine. However, the increase in GLP-1 observed after gastric bypass surgery can be partly attributed to increased delivery of nutrients to the distal parts of the intestine with a greater L-cell density [27,28,29,30,32]. In this short communication, we will outline the present knowledge on the nutrient-induced secretion of GLP-1, focusing on the macronutrients and their building blocks.

## 2. Carbohydrate-Induced GLP-1 Secretion

Carbohydrates are absorbed as monosaccharides in the small intestine, and therefore, complex carbohydrates require digestion by salivary and gastric enzymes prior to absorption. As the glucose-induced secretion of GLP-1 is one of the essential requirements of the hormone’s incretin status [16,17,20], the molecular mechanism of this has been an area of interest since the first discoveries of the hormone. In 1995, it was demonstrated that sucrose could stimulate GLP-1 secretion through both an early mechanism and a subsequent mechanism involving luminal contact [36]. In a study on the perfused rat ileum 2 years later, a Na^+^/glucose transporter and a Na^+^-independent fructose transporter were suggested as luminal mediators of monosaccharide-induced GLP-1 secretion [37]. Since these findings, studies on both primary L-cells and GLUTag cells (immortal GLP-1-secreting cells derived from a mouse colon) have proven that single L-cells respond to glucose and fructose, supporting the hypothesis of a sensory mechanism for monosaccharide-induced GLP-1 secretion [38,39,40].

In a study on GLUTag cells, low glucose concentrations (0–25 mmol/L) have an increased membrane depolarization and action potential frequency [38]. This depolarizing effect was diminished by the K_ATP_ channel opener diazoxide and the metabolic inhibitor Na^+^-azide, supporting the metabolism of glucose and subsequent closure of the K_ATP_ channel as potential pathways for glucose-induced GLP-1 secretion at low glucose concentrations [38] (Figure 1). In agreement with this, the K_ATP_ channel blocker tolbutamide was demonstrated to stimulate GLP-1 secretion, and both the K_ATP_ channel subunits and glucokinase could be detected in GLUTag cells through real-time PCR [38]. In the same cell line, different concentrations of fructose were found to stimulate GLP-1 secretion via closure of K_ATP_ channels [39]. In contrast to data obtained with low-concentration glucose (which support the involvement of intracellular metabolism and closure of K_ATP_ channels in glucose-induced GLP-1 secretion), a high concentration (100 mmol/L) of the non-metabolizable glucose analog methyl-α-glucopyranoside has been found to stimulate GLP-1 secretion and depolarize the GLUTag cell through an inward current [39]. This effect of methyl-α-glucopyranoside was abolished in the absence of Na^+^ ions and in the presence of phloridzin (an inhibitor of the Na^+^/glucose transporter). In further support, the Na^+^/glucose transporters SGLT1 and SGLT3 could be detected in the GLUTag cells by RT-PCR, supporting a role of these Na^+^/glucose transporters in GLP-1 secretion induced by methyl-α-glucopyranoside, at least in GLUTag cells [39]. In the same study, GLP-1 secretion stimulated by a low glucose concentration (0.5 mmol/L) was decreased (≈40%) in the presence of phloridzin, suggesting that glucose stimulates GLP-1 secretion at least partly through SGLT-mediated uptake [39] (Figure 1).

Supporting the GLUTag cell line studies, it has been demonstrated that primary mouse L-cells also respond to stimulation with glucose, methyl-α-glucopyranoside, and tolbutamide [40], and several in vivo studies support the findings of SGLT1 as a mediator of monosaccharide-induced GLP-1 secretion [41,42,43] (Figure 1). In a study of mice, glucose and methyl-α-glucopyranoside both stimulated GLP-1 secretion after intra-intestinal administration, an effect that was abolished when the compounds were administered in combination with phloridzin [41]. In a different study, *Sglt1*^−/−^ mice had no GLP-1 response 5 min after oral administration of glucose (6 mg/g body weight), whereas a significant increase in the total systemic GLP-1 was observed for their wildtype littermates after the same treatment [42]. Similarly, in an isolated perfused rat small intestine model, SGLT1 has also been demonstrated to be crucial for glucose-induced GLP-1 secretion [43]. In this model, GLP-1 secretion was induced by both glucose and methyl-α-glucopyranoside, and again, the effect was abolished when the secretagogues were administered in combination with phloridzin or in the absence of Na^+^ [43]. This again supports the importance of SGLT1 in glucose-induced GLP-1 secretion in rodents. Interestingly, when administered at the same concentration (20% *w*/*v*), glucose induced a greater GLP-1 response than methyl-α-glucopyranoside, implying that an additional mechanism potentiates the glucose-induced GLP-1 response at high concentrations of glucose. The authors suggested GLUT2-mediated uptake and intracellular metabolism of glucose, leading to the opening of K_ATP_ channels, as a possible mechanism [43] (Figure 1). This hypothesis is supported by the observation that the blockage of GLUT2 decreased, but did not abolish, the glucose-induced GLP-1 secretion and that secretion could be induced by tolbutamide (causing closure of the K_ATP_ channels and depolarization) [43], similar to the results obtained in the in vitro studies referenced earlier [38,40].

Whereas there are strong indications for an essential role of SGLT-1 and K_ATP_ channels in in vitro and in vivo animal models, sufficient data from humans are missing. In one study from 2017, SGLT-1 and GLUT2 appeared to be important for the glucose-induced GLP-1 secretion in isolated human ileal mucosal explants, whereas the role of the K_ATP_ channels was less clear [44]. In the human ileal isolates, a high concentration of glucose (300 mmol/L) stimulated GLP-1 secretion, and this effect was blocked in the absence of Na^+^ or when glucose was co-administered with phloridzin or the GLUT2 inhibitor phloretin [44]. Regarding the role of K_ATP_ channel activity, tolbutamide (500 µmol/L) did not stimulate GLP-1 secretion, although the effect of glucose was abolished in the presence of the K_ATP_ channel opener diazoxide or the ATP synthesis inhibitor 2,4-DNP [44] (Figure 1). As Sun et al. suggested, a possible explanation for this inconsistency is that diazoxide simply counteracts the depolarization from SGLT1-mediated Na^+^/glucose transport and that 2,4-DNP blocks a metabolism-dependent effect of GLUT2-mediated glucose uptake, which is not related to K_ATP_ channel activity [44]. Although the study demonstrated that glucose can directly stimulate human L-cells, important limitations must be considered. As the study was conducted on isolated tissue, the cells were not polarized as under normal physiological conditions. Moreover, ileal mucosa is very unlikely to encounter the glucose concentrations of the study under normal conditions, as most glucose would be absorbed in the duodenum. Thus, whether the findings on the molecular mechanism of glucose-induced GLP-1 secretion from rodent and cell studies are directly transferrable to humans is not clear. Consequently, although the mechanisms of carbohydrate-induced GLP-1 secretion are well-described in the literature, and a consensus on the mechanism in experimental models exists, essential pieces are still missing in the puzzle of transferability.

In addition to digestible carbohydrates, microbial fermentation products of non-digestible carbohydrates have gained interest as potential GLP-1 secretagogues. An important example is short-chain fatty acids (SCFAs). In vitro and in vivo studies of rodent models have demonstrated that SCFAs may mediate GLP-1 secretion through the free fatty acid receptors FFA2 and FFA3 [45,46] (Figure 1). However, in a study on a perfused rat colon, the importance of FFA2 and FFA3 in SCFA-mediated GLP-1 secretion was questioned [47]. In this model, SCFAs affect the colonic GLP-1 secretion but not through FFA2 or FFA3, as neither agonizing nor antagonizing the receptors influences GLP-1 secretion in a perfused rat colon. Instead, SCFAs seem to increase GLP-1 secretion through intracellular metabolism, possibly followed by the closure of ATP-sensitive potassium channels [47] (Figure 1). In two comparable human studies on overweight and obese individuals, colonic infusions of SCFAs positively impacted fat oxidation, energy expenditure, and PYY levels [48,49]. In contrast, no significant increase in GLP-1 was observed. Again, therefore, it remains to be seen if the SCFA-mediated GLP-1 secretion observed in some animal models is transferrable to humans.

## 3. Fat-Induced GLP-1 Secretion

Fat is ingested in the form of triglycerides, phospholipids, or cholesterol, and the absorption of dietary fat is a far more complicated process than the absorption of dietary carbohydrates and proteins. In the intestinal lumen, bile salts are responsible for emulsifying the lipids, making them accessible for digestion by pancreatic lipases. The breakdown of triglycerides yields 2-monoacylglycerol (2-MAG) and free fatty acids (FFAs), whereas the breakdown of the most abundant phospholipid, phosphatidylcholine, yields lysophosphatidylcholine and FFAs. The digestive products join with bile acids to form amphiphilic micelles, which facilitate the absorption of the breakdown products. Within the enterocytes, the lipids are re-esterified and incorporated into lipoproteins termed chylomicrons, which are released through the lymphatic system [50]. It is well-established that the ingestion of dietary fat induces a significant GLP-1 response in humans [51,52]. As the pathway of absorption is complex and involves the generation of several intermediate products, various potential mediators of fat-induced GLP-1 secretion have been investigated, and some were also identified. Several studies have demonstrated that the level of lipid-induced GLP-1 secretion is reduced in the presence of the lipase inhibitor orlistat [53,54,55,56], suggesting that the enzymatic digestion of lipids is crucial for stimulating GLP-1 secretion.

One digestive product known to stimulate GLP-1 secretion is FFAs. However, important differences between FFAs of distinct types have been demonstrated with respect to stimulatory potential. In humans, the ingestion of monounsaturated fat induces a greater GLP-1 response than ingestion of polyunsaturated or saturated fat [57,58]. Similarly, chain length seems to be a determining factor for the ability of FFAs to stimulate GLP-1 secretion, although it is not possible to define the exact chain length required for inducing a GLP-1 response from the existing literature. In fetal rat cell cultures, incubation with mono-unsaturated FFAs with a chain length of 16 and 18 carbon atoms stimulated GLP-1 secretion, whereas incubation with mono-unsaturated FFAs with a chain length of 14 carbon atoms was unable to induce a response [59]. In humans, duodenal infusion with a saturated FFA with a chain length of 12 carbon atoms stimulated GLP-1 secretion, whereas infusion with a saturated FFA with a chain length of 10 carbon atoms was ineffective [60]. Although the exact chain length required for stimulation is unclear, the studies indicate that longer FFAs are more powerful intestinal stimulators than shorter FFAs. A proposed explanation for this difference is the formation of chylomicrons [60], as long FFAs are incorporated in chylomicrons and transported in the lymph following absorption, whereas short FFAs are transported in the portal system [61]. However, this hypothesis is only partly supported by observations from in vitro and in vivo studies [62,63]. In a study on GLUTag cells, primary duodenal murine cells and primary duodenal human cells, physiological concentrations of chylomicrons stimulated GLP-1 secretion [63]. In GLUTag cells, the chylomicron-induced GLP-1 secretion involved lipolysis of the chylomicrons and activation of the FFA receptor FFA1 (Figure 2), whereas the mechanism in the primary cells was unclear [63]. In a study on rats, the chylomicron inhibitor L-81 delayed the fat-induced GLP-1 response, measured in the lymph after duodenal infusion of fat alone or fat in combination with L-81 [62] (Figure 2). However, during a 6-h period, the GLP-1 output was still greater in the L-81 group when compared with a saline control group [62]. The delayed response observed when preventing chylomicron formation could indicate that chylomicrons are important for an early fat-induced GLP-1 response, whereas other mechanisms are important for the later response [62]. As fat-induced GIP secretion was completely abolished in the presence of L-81, the delayed GLP-1 response may also just reflect the absence of GIP-induced GLP-1 secretion, questioning if chylomicrons have a direct effect on GLP-1 secretion in vivo [62].

Despite FFA1 being excluded as a mediator of chylomicron-induced GLP-1 secretion in primary duodenal murine cells and primary duodenal human cells [63], other models have proposed a role for FFA1 in FFA-induced GLP-1 secretion [64,65]. In a study from 2008, the co-expression of FFA1 and GLP-1 was demonstrated in enteroendocrine cells in mice [64]. In the same study, the *FFA1^lacZ/lacZ^* mice had a significantly lower GLP-1 plasma level 60 min after acute oral fat administration when compared with their wildtype littermates [64]. Supporting this, basolateral activation of FFA1 with both an endogenous ligand (linoleic acid) and synthetic agonists increased GLP-1 secretion in an isolated perfused rat small intestine model [65]. Importantly, this effect was not observed when the ligands were infused luminally, suggesting that FFAs act on FFA1 following absorption [65]. Adding to the complexity of lipid-induced GLP-1 secretion, a study from 2014 demonstrated that agonists acting on the G_q_ signaling pathway of FFA1, which is the pathway for FFA-induced activation of the receptor, induced a lower GLP-1 response following oral administration in mice when compared with agonists acting on both the G_q_ and G_s_ pathways [66]. In a phase I clinical trial, the FFA1 agonist TAK-875 did not have a noticeable effect on GLP-1 secretion when administered in therapeutic concentrations [67]. Instead, the improved glycemic controls in subjects receiving the drug was attributed to its effects on FFA1 in pancreatic beta cells, stimulating insulin secretion [67]. As an important side note, the development of the drug was terminated in the phase III trials due to concerns over liver toxicity not related to FFA1 [68]. Thus, the activation of FFA1 by FFAs is probably not sufficient to induce a pronounced incretin response on its own.

Following a fat-containing meal, FFAs are not the only component of dietary fat known to stimulate GLP-1 secretion. GPR119 is a well-recognized mediator of GLP-1 secretion in both rodents and humans, where the receptor acts through interactions with lipid derivatives such as lysophosphatidylcholine [69], oleoylethanolamide [70], and 2-monoacylglycerols [71,72] (Figure 2). In vivo studies on different knock-out mice strains have indicated that GPR119 is more important for GLP-1 secretion than FFA1 and that the two receptors work in synergy to create a lipid-induced GLP-1 secretion [72]. In rodents, a GRP119 agonist has been demonstrated to increase the GLP-1, GIP, and PYY levels [73], making this receptor an attractive therapeutic target. Unfortunately, the results from studies assessing the potential of GPR119 agonists in type 2 diabetes therapy have been disappointing. Many pharmaceutical companies have failed in the attempt to develop a GPR119 agonist, and no drug candidate has passed the phase II clinical trials so far, with one challenge being the required lipophilic properties of the drugs [74,75]. Thus, although attempts to uncover molecular pathways of lipid-induced GLP-1 secretion have been successful, the discoveries have yet to be translated into relevant therapy. Hopefully, this issue can be solved through focused research in the future.

## 4. Protein-Induced GLP-1 Secretion

Protein is absorbed as peptides or amino acids in the small intestine, and thus, the complex proteins from a diet require enzymatic digestion prior to absorption. A high-protein diet has been demonstrated to improve weight loss and weight maintenance when compared with a low-protein diet [76,77], although the underlying mechanisms are not entirely clear. Some studies indicated that dietary protein is more satiating than carbohydrates or fat [78,79,80,81], while others did not find a significant difference between the three macronutrients [82,83]. A similar discrepancy applies to the thermogenic effect and the effect on plasma GLP-1 levels when comparing proteins with carbohydrates and fat. Reports on both a higher [80] and a similar [83] thermogenic effect of protein exist. Early studies showed that ingestion of a protein meal or free amino acids induces a transient increase in plasma GLP-1 levels [51,52]. However, whether this increase is greater in magnitude than the increase induced by carbohydrates or fat is unclear, as the conclusions from different studies are inconsistent. In a randomized three-way crossover design study, no difference in plasma GLP-1 was observed in healthy or obese males after consumption of an isocaloric meal with a high proportion of either protein, carbohydrates, or fat [81]. On the contrary, other studies have demonstrated a greater GLP-1 response following a high-protein meal when compared with isocaloric meals lower in protein [83,84]. As the test meals within all reported studies were isocaloric, different energy contents of the meals cannot explain the different results. Volume differences in the meals may have influenced the GLP-1 secretion (as a meal with a higher volume would stay in the intestinal system for longer). In the one study that did not demonstrate a difference, all test meals consisted of pasta (normal or high-protein) and tomato sauce supplemented with protein, carbohydrates, or fat [81]. In contrast, the meal size varied more in the two studies demonstrating a difference [83,84]. Moreover, different types of protein may stimulate GLP-1 secretion with different potencies. As such, the source of the protein should also be considered when comparing the effect of protein on GLP-1 secretion.

As is the case for carbohydrates and fat, different digestive products of dietary protein are thought to stimulate GLP-1 secretion through different mechanisms. Peptones (protein hydrolysates of an animal source) have been demonstrated as a potent stimulator of GLP-1 secretion in murine primary colonic L-cells, STC-1 cells, and the isolated perfused rat small intestine model [85,86,87]. Peptide Transporter 1 (Pept1) and the G-protein-coupled Calcium-Sensing Receptor (CaSR) have both been suggested as potential mediators of peptone-induced GLP-1 secretion [85,87] (Figure 3). However, the importance of each of these receptors is not consistent between the different models. In murine primary colonic L-cells, the inhibition of Pept1 with 4-aminomethylbenzoic acid (4-AMBA) did not abolish the peptone-induced GLP-1 secretion, neither for low nor high peptone concentrations (5 or 50 mg/mL). In contrast, the blockage of Pept1 with the same inhibitor did abolish the GLP-1 response to luminally infused peptones (50 mg/mL) in the isolated perfused rat small intestine [87] (Figure 3). An essential difference between the two models is the preserved cellular polarization in the perfused rat intestine, which is lost in the cellular model. Thus, these results could indicate that, in a physiological setting, peptones require absorption to elicit a GLP-1 response. In a cellular model, this may not be important since the peptones can interact directly with the mediators normally located on the basolateral side of the cell. Supporting this, inhibiting the amino acid-sensing receptor CaSR with the negative allosteric modulator NPS2143 diminished the peptone-induced GLP-1 secretion in both murine colonic L-cells and after vascular stimulation in the perfused rat small intestine [85,87] (Figure 3). As the positive allosteric modulator calindol only stimulated GLP-1 secretion when infused vascularly in the perfused rat small intestine, CaSR must be located on the basolateral side of the L-cell [87] (Figure 3). Thus, peptones may stimulate GLP-1 secretion through absorption by Pept1, followed by intracellular digestion to free amino acids, finally activating CaSR on the basolateral side of the cell.

Pept1 and CaSR are not the only mediators associated with protein-induced GLP-1 secretion. Whereas peptones are a non-specific mix of different peptides and free amino acids, specific peptides and free amino acids have been associated with distinct molecular pathways of GLP-1 secretion. Similar to peptones, dipeptides such as glycine-sarcosine (Gly-Sar) and glycine-glycine (Gly-Gly) have been demonstrated to stimulate GLP-1 secretion in murine primary colonic L-cells, STC-1 cells, and in the isolated perfused rat small intestine [85,87,88]. In contrast to the peptone-induced GLP-1 secretion, inhibition of Pept1 with 4-AMBA did abolish GLP-1 secretion induced by the non-metabolizable Gly-Sar in murine primary colonic L-cells [85]. Similar results were obtained for Gly-Gly in STC1 cells [88] and for Gly-Sar in the perfused rat small intestine [87]. This indicates a direct mechanism for dipeptide-induced GLP-1 secretion after uptake by Pept1, independent of intracellular digestion to free amino acids and interaction with CaSR. As Pept1 is a proton-coupled transporter, this mechanism may involve membrane depolarization and a subsequent influx of Ca^2+^ from voltage-gated Ca^2+^ channels. In support of this, GLP-1 secretion mediated by Gly-Sar and Gly-Gly was diminished by the presence of the Ca^2+^ channel inhibitor nifedipine in vitro [85,88] (Figure 3). However, this hypothesis was not supported by results from the perfused rat small intestine model, where GLP-1 secretion induced by Gly-Sar was not abolished in the presence of nifedipine [87]. Thus, the exact pathway coupling dipeptides and Pept1 to GLP-1 secretion remains to be clarified.

In a study on female individuals with obesity, the circulating levels of eight specific amino acids (isoleucine, leucine, lysine, methionine, phenylalanine, proline, tyrosine, and valine) were positively correlated with the plasma GLP-1 levels [89], and in experimental models, a range of specific free amino acids has been demonstrated to induce GLP-1 secretion with different potencies and through different mechanisms [90,91,92,93,94,95]. From these studies, the most important amino acids for GLP-1 secretion appear to be L-glutamine, L-tryptophan, L-phenylalanine, L-asparagine, L-arginine, and L-valine. In murine primary colonic L-cells and GLUTag-cells, glutamine-induced GLP-1 secretion involves a rise in cytosolic Ca^2+^ and requires the presence of Na^+^ in the media [90,91], suggesting Na^+^-coupled uptake and the subsequent opening of voltage-gated Ca^2+^ channels as a potential pathway for glutamine-induced GLP-1 secretion (Figure 3). However, the exact transporters involved remain to be established. An additional mechanism proposed for glutamine-induced GLP-1 secretion is an elevation of cAMP [91], although the results from GLUTag cells are somewhat conflicting. In one experimental set-up, the protein kinase A (PKA) inhibitor H89 did not affect glutamine-induced GLP-1 secretion [90], whereas monitoring the cAMP levels in another experimental design revealed that glutamine increased the cAMP levels in GLUTag cells [91]. Thus, glutamine may increase GLP-1 through a pathway downstream of increased cytosolic cAMP, independent on PKA activation (Figure 3). Importantly, glutamine has been demonstrated to increase GLP-1 and insulin levels following oral administration in humans [96], proposing a therapeutic potential of the amino acid.

As for peptone-induced GLP-1 secretion, CaSR has also been linked to the GLP-1 secretion induced by certain free amino acids [93,95] (Figure 3). In the perfused rat small intestine, inhibition of CaSR decreased the GLP-1 response to the luminal infusion of vamin (a mix of 18 different amino acids) [95]. However, inhibition of CaSR did not affect the GLP-1 response to vascular-infused arginine [95], suggesting that CaSR is only involved in the GLP-1 response to some amino acids. One amino acid where CaSR does seem to be involved in the GLP-1 response is phenylalanine. In the perfused rat small intestine, phenylalanine has been demonstrated to stimulate GLP-1 secretion following both luminal and vascular infusion [95], and as the amino acid is efficiently absorbed from the rat intestine [95], a basolateral mechanism involving CaSR is plausible. This is supported by another rodent study where intra-ileal administration of phenylalanine reduced the acute food intake in rats, an effect that was abolished when co-administered with the CaSR inhibitor NPS2143 [93]. In the same study, phenylalanine-induced GLP-1 secretion from different cell lines was also diminished in the presence of NPS2143, supporting a role for CaSR in phenylalanine-induced GLP-1 secretion [93]. Two additional G-protein coupled receptors, GPRC6A and GPR142, have also been associated with GLP-1 secretion induced by specific amino acids [92,94] (Figure 3). In the GLUTag cells, L-ornithine-induced GLP-1 secretion was significantly reduced when the cells were treated with siRNA for GPRC6A [92]. However, in the GPRC6A knock-out mice, no difference in plasma GLP-1 was observed 15 min after oral administration of L-arginine or L-ornithine when compared to the wildtype littermates [97]. Although the GLP-1 response was attenuated 30 and 60 min after administration of the amino acids, this did not significantly impact the total GLP-1 release [97]. Adding to this, GPRC6A could not be detected in cells from segments of the mice’s duodenum, jejunum, or ileum [95], again challenging the idea of GPRC6A as an important mediator of amino acid-induced GLP-1 secretion.

Altogether, unidentified mechanisms remain to be uncovered to obtain a complete understanding of protein-induced GLP-1 secretion. Glutamine and arginine have been demonstrated to induce GLP-1 secretion in humans [96,98], and recently, valine was demonstrated as a potent stimulator of GLP-1 secretion when infused luminally in the perfused rat small intestine [95]. These findings strongly emphasize the relevance of uncovering the mechanisms of protein-induced GLP-1 secretion, as these may prove valuable in future therapy.

## 5. Future Perspectives

Through a joined effort and various experimental models, recent research has begun to reveal the complexity of nutrient-induced GLP-1 secretion. It turns out that the digestion products of the three macronutrients stimulate secretion through widely different mechanisms, with a large number of sensors and receptors involved in the nutrient-induced GLP-1 secretion. As we dive further into the mechanisms involved, we may be able to utilize approaches involving stimulation of the secretion of endogenous gut hormones in the clinic, supporting and expanding existing therapeutic options. An important feature of new GLP-1 based therapeutics is to reduce side effects. Targeting endogenous secretion is a potential approach, and perhaps in combination with incretin enhancers like DPP4 inhibitors or somatostatin subtype 5 receptor antagonists (preventing feedback inhibition of GLP-1 secretion by somatostatin), this may be even more effective than current GLP-1-based therapies. However, we still have a lot to learn and have probably only begun to uncover the mechanistic links between eating and hormone secretion. In several of the studies on GLP-1 secretion, it is curious that the timing rather than the total hormone secretion is affected by the nutrient stimulus. The implication of this is difficult to determine. The effectiveness of gastric bypass operations (which induce a high and transient elevation in GLP-1) supports timing as the main factor, whereas the superior effect of the weekly GLP-1Ras compared with the short-acting agonists supports total hormone secretion. As stimulating the endogenous GLP-1 secretion of the L-cells is an attempt to mimic gastric surgery, the effectiveness of the strategy would rely on the timing as a determining factor.

An important challenge in the field is the sometimes-limited transferability between experimental animal models and human subjects. This has been exemplified in several studies on secretion elicited experimentally with lipids and amino acids, where attempts at clinical application have been disappointing. To obtain a better understanding of the nutrient-induced GLP-1 secretion and utilize this understanding in therapy, we need more human studies and more experimental studies with improved clinically relevant models to confirm the patterns that we are beginning to understand from in vitro and in vivo animal models. One attempt to improve preclinical models is the use of human intestinal organoids, which display a high degree of similarity with human intestinal cells [99]. Hopefully, these attempts and the following years of research will provide us with some of the missing pieces of knowledge that exist for all the macronutrients.

## Figures and Tables

**Figure 1 metabolites-12-00420-f001:**
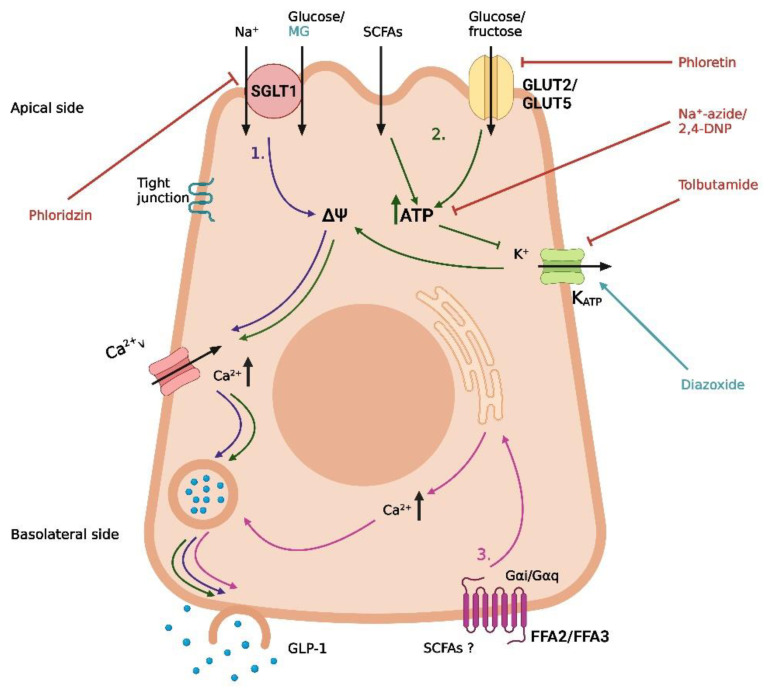
Proposed mechanisms of carbohydrate-induced GLP-1 secretion in intestinal L-cells. (1) Uptake of glucose/Na^+^ by SGLT1, leading to depolarization and opening of voltage-gated Ca^2+^-channels. (2) GLUT2-mediated uptake of glucose, GLUT5-mediated uptake of fructose, and absorption of short-chain fatty acids (SCFAs), followed by intracellular metabolism, closure of K_ATP_ channels, depolarization, and opening of voltage-gated Ca^2+^-channels. (3) Basolateral activation of FFA2 or FFA3 by SCFAs, leading to the release of intracellular deposits of Ca^2+^. Methyl-α-glucopyranoside (MG) is an agonist of SGLT1, phloridzin is an SGLT-1 inhibitor, phloretin is a GLUT2 inhibitor, Na^+^-azide and 2,4-DNP are metabolic inhibitors, tolbutamide is a K_ATP_ channel inhibitor, and diazoxide is a K_ATP_ channel opener. The graphic was created with BioRender.com.

**Figure 2 metabolites-12-00420-f002:**
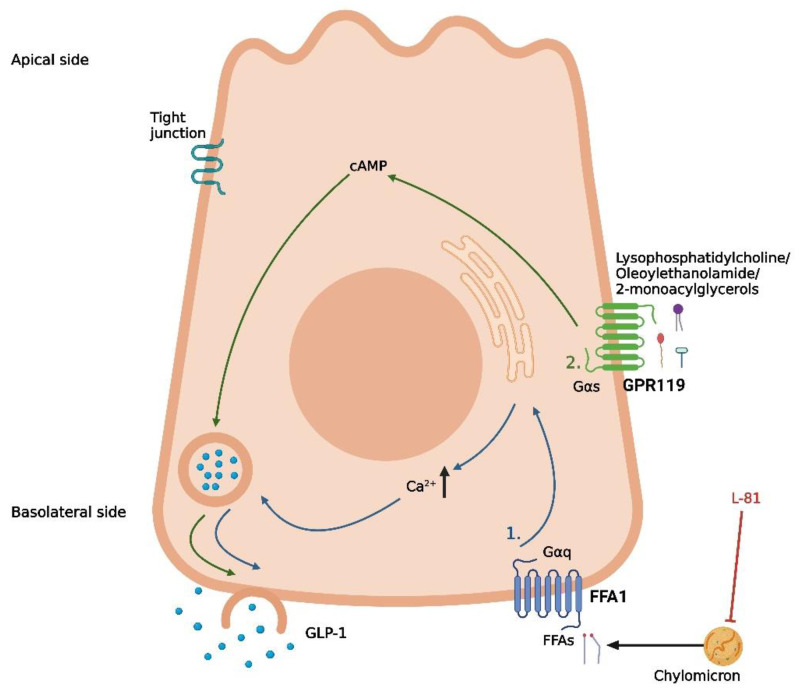
Proposed mechanisms of fat-induced GLP-1 secretion in intestinal L-cells. (1) Lipolysis of absorbed chylomicrons, followed by the activation of FFA1 by free fatty acids (FFAs), leading to the release of intracellular deposits of Ca^2+^. (2) Basolateral activation of GPR119 by lipid derivatives (lysophosphatidylcholine, oleoylethanolamide, and 2-monoacylglycerols), leading to an increase in cAMP. L-81 is a chylomicron inhibitor. The graphic was created with BioRender.com.

**Figure 3 metabolites-12-00420-f003:**
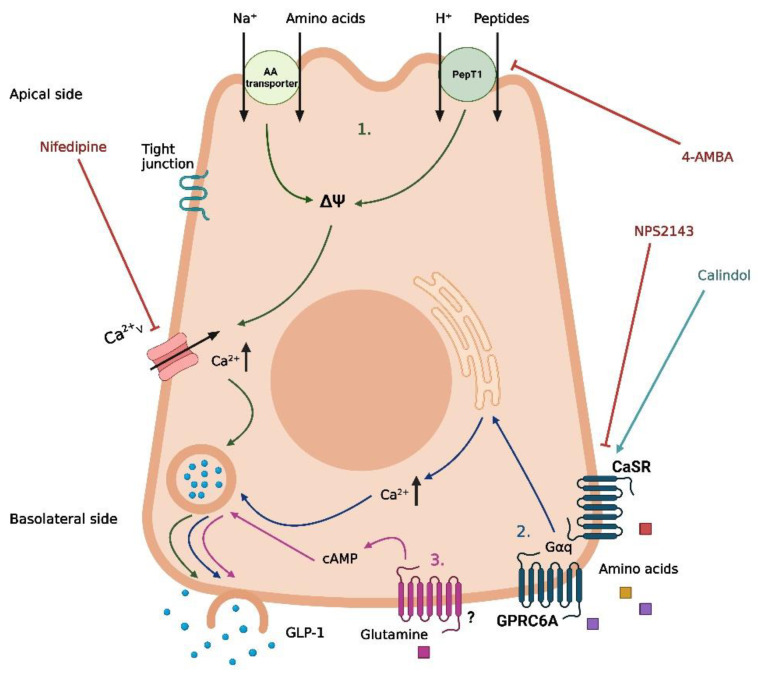
Proposed mechanisms of protein-induced GLP-1 secretion in intestinal L-cells. (1) Uptake of free amino acids through coupled transport with Na^+^ and uptake of peptides/H^+^ by PepT1, leading to depolarization and opening of voltage-gated Ca^2+^-channels. (2) Basolateral activation of CaSR and GPRC6A by free amino acids following absorption, leading to the release of intracellular deposits of Ca^2+^. (3) Basolateral activation of an unknown receptor by glutamine following absorption, leading to an increase in cAMP. Nifedipine is a Ca^2+^ channel inhibitor, 4-aminomethylbenzoic acid (4-AMBA) is a PepT1 inhibitor, NPS2143 is a CaSR inhibitor, and calindol is a CaSR agonist. The graphic was created with BioRender.com.

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
