# Peer review of "The Sensory Mechanisms of Nutrient-Induced GLP-1 Secretion"

_metabolites, 2022, doi:10.3390/metabo12050420_

Round 1

Reviewer 1 Report

The authors present a concise review of preclinical and clinical evidence for mechanisms underlying the nutrient-mediated induction of GLP-1. This paper is overall well organized, and the data are presented clearly, although grammatical and syntax errors throughout are a bit distracting. I think this review would benefit from provision of greater clinical context in the introduction and inclusion of figures to illustrate the pathways highlighted in the text. My specific comments are as follows:

  1. Inclusion of a figure with the anatomic sites of EECs would be very helpful, including the various locations of L-cells. Do nutrients differentially regulate GLP-1 secretion in these various locations (i.e. ileum versus colon)? 
  2. The final paragraph of the introduction suggests the clinical relevance of this review, but the potential impact of this line of research could be developed more. Are there shortcomings of GLP1RA therapy that specifically could be addressed through nutrient-based strategies or other therapeutics that increase endogenous GLP-1 instead? Providing greater overarching context of drug development would be helpful and underscore the clinical relevance of the presented data. Similarly, in the section describing failed efforts thus far to develop GPR119 agonists for type 2 diabetes treatment, are there theoretical advantages of GPR119 agonists over GLP1RAs? Do any of these pathways also regulate the secretion of other incretins?
  3. A figure depicting the potential intracellular pathways regulating GLP-1 secretion would be extremely helpful. Inclusion of the reagents used in many of the in vitro studies cited would also aid the reader.
  4. The authors note in the concluding paragraphs that preclinical models should more closely simulate human physiology. Can the authors suggest specific strategies to accomplish this? A description of the key features that would characterize an optimal new GLP-1-based therapeutic would be of interest.
  5. In several of the experimental models, the timing rather than total amount of GLP-1 secretion is affected by the intervention. Is it known how the slowed/accelerated release of GLP1 impacts its biological effects? 

Reviewer 2 Report

The authors reviewed nutrient-induced GLP-1 secretion. A review of the relationship between diabetes and "nutrient induced GLP-1 secretion" is needed. Also, it is necessary to review the relationship of gut-brain-pancratic axis and nutrients to the synthesis and secretion of GLP-1. The effects of non-digestible carbohydrates and short-chain fatty acids on GLP-1 as well as GLP-1 secretion by carbohydrates and monosaccharides need to be reviewed. A review of GLP-1 secretion by mixed-nutrient foods is also needed. In addition, a table or figure is required for each mechanism of GLP-1 secretion.

Round 2

Reviewer 2 Report

Authors reviewed the sensory mechanisms of nutrient-induced GLP-1 secretion. This is a well-reviewed paper and each section is well organized.